# Positive Psychology Interventions in Early-Stage Cognitive Decline Related to Dementia: A Systematic Review of Cognitive and Brain Functioning Outcomes of Mindfulness Interventions

**DOI:** 10.3390/brainsci15060580

**Published:** 2025-05-28

**Authors:** Dimitra Vasileiou, Despina Moraitou, Konstantinos Diamantaras, Vasileios Papaliagkas, Christos Pezirkianidis, Magda Tsolaki

**Affiliations:** 1Laboratory of Psychology, Department of Cognition, Brain and Behavior, School of Psychology, Aristotle University of Thessaloniki (AUTh), 54124 Thessaloniki, Greece; dimitrvasileiou@gmail.com (D.V.); demorait@psy.auth.gr (D.M.); 2Laboratory of Neurodegenerative Diseases, Center for Interdisciplinary Research and Innovation (CIRI), Aristotle University of Thessaloniki (AUTh), 54124 Thessaloniki, Greece; tsolakim1@gmail.com; 3Department of Information & Electronic Engineering, International Hellenic University, 57400 Sindos, Greece; kdiamant@ihu.gr; 4Department of Biomedical Sciences, International Hellenic University, 57400 Thessaloniki, Greece; 5Laboratory of Positive Psychology, Panteion University of Social & Political Sciences, 17671 Athens, Greece; christospez@hotmail.com; 6Greek Association of Alzheimer’s Disease and Related Disorders (GAADRD), 54643 Thessaloniki, Greece

**Keywords:** non-pharmacological interventions for cognitive decline related to dementia, positive psychology interventions, mindfulness, character strengths, SCD, MCI, mild AD dementia

## Abstract

**Background**: Dementia is a global condition affecting over 55 million people. Since there is no treatment, non-pharmacological interventions aim to delay its progression in a safe and cost-effective way. The extant literature suggests that Positive Psychology Interventions (PPIs) can probably be effective for this purpose. The systematic review aims to assess the effectiveness of PPIs as non-pharmacological interventions for mild cognitive decline related to dementia by evaluating their effectiveness in cognitive functions and brain functioning in people with Subjective Cognitive Decline (SCD), Mild Cognitive Impairment (MCI), and mild Alzheimer’s disease dementia (AD). **Methods**: A comprehensive search conducted in the databases Scopus, PubMed, ScienceDirect and PsychINFO (December 2024–March 2025) published between 2015 and 2025 to identify records that met inclusion criteria: studies included patients with SCD, MCI and mild AD dementia, implemented PPIs, Randomized controlled trials (RCTs) and pre–post intervention studies with measurable outcomes, assess at least one of the following: cognitive functions and brain functioning. **Results**: The systematic review included 12 studies (N = 669 participants) that can answer the research question. Only mindfulness interventions were identified. Findings suggest that different types of mindfulness interventions, such as the Mindfulness Awareness Program (MAP) and Mindfulness Training (MT), may be efficient for improving specific cognitive functions (e.g., working memory and attention) and influencing biological pathways related to cognitive decline. However, long-term efficacy has not been demonstrated, and results are mixed and unclear. **Conclusions**: Μindfulness interventions seem promising for enhancing cognition and brain functioning in older adults with cognitive decline, although the data is limited. However, limitations such as the heterogeneity of the studies and the diversity of the interventions make it necessary for more systematic and organized research to be conducted on the implementation of such interventions. At the same time, it is proposed to examine the effectiveness of other constructs of positive psychology, such as character strengths (CS).

## 1. Introduction

The purpose of this research is to investigate the potential of positive psychology interventions (PPIs) to enhance cognitive function and brain functioning in patients experiencing early-stage cognitive decline due to dementia. Evidence suggests that elements of positive psychology, such as mindfulness and character strengths, can improve cognitive and neurological aspects [1,2]. Therefore, PPIs may offer a non-pharmacological approach to support individuals with dementia. This review aims to identify and evaluate existing studies that have applied and evaluated interventions focusing on positive psychology constructs in individuals with early-stage cognitive decline related to dementia. The existing literature indicates correlations between positive psychology constructs and cognitive and brain functioning [1,2]. However, the application of these constructs in patients with cognitive decline related to dementia—particularly as non-pharmacological interventions—remains limited despite promising indications of their potential effectiveness. The following sections will analyze the need for novel dementia interventions and review the literature supporting the potential of PPIs.

Dementia is a widespread condition that affects over 55 million cases globally, with approximately 10 million new patients each year. It is estimated that by 2050, the number of people living with dementia will rise to 139 million [3]. Dementia’s impact is significant, not only for those living with dementia but also for their families and the community [3,4].

Cognitive decline related to dementia often occurs along a spectrum, beginning with Subjective Cognitive Decline (SCD), where individuals report a self-perceived decline in their cognitive abilities, particularly in memory. At this stage, the decline is primarily assessed through self-reports [5,6,7,8,9], and it is very difficult to detect using neuropsychological assessments due to the very mild nature of the symptoms [8,10,11,12,13]. These symptoms are usually indicators of objective cognitive impairment in the future. Mild Cognitive Impairment (MCI) is typically the next stage of cognitive decline, characterized by measurable deficits in cognitive functions, while daily functioning remains largely intact [14,15]. MCI is classified into two subtypes: amnestic MCI, which primarily involves memory deficits, and non-amnestic MCI, which primarily affects other cognitive functions [13]. Patients with MCI, especially those with amnestic subtypes, are at higher risk of developing AD dementia, although it is not necessary [15,16]. Mild AD dementia represents the early stage of AD dementia, during which cognitive deficits are more apparent. Patients are still able to be involved in their daily routines but may require assistance with more complex tasks [17].

While there is no cure for dementia, non-pharmacological interventions are often the first choice because they are generally safer and less costly than pharmacological treatments, which also tend to be less effective. Additionally, non-pharmacological interventions can be tailored to meet each patient’s needs [18,19,20]. It is well-established that environmental and modifiable factors play a significant role in delaying the progression of dementia symptoms. Such factors include educational level, smoking, physical exercise, depression, social isolation and more. Non-pharmacological interventions to date have focused on strengthening factors such as nutrition, cardiovascular health, physical activity, cognitive training and social engagement. However, current interventions do not specifically address psychological factors such as trait neuroticism, negative thoughts, and positive traits and behavior [4,20].

Positive psychology is a field of psychology that shifts the focus from psychopathology to psychological flourishing and the enhancement of positive character constructs. Its purpose is not only to address dysfunctional situations but also to cultivate positive aspects of life [21]. Key constructs within positive psychology include Character Strengths (CS), which are positive traits of character and mindfulness, which is the process of focusing on the present moment while utilizing cognitive functions such as attention and awareness [22]. These constructs are closely interlinked; one would be incomplete without the other. Practicing mindfulness can enhance CS and vice versa; practicing and cultivating CS can deepen mindfulness [23]. PPIs have been applied to older adults, although to a lesser extent than to younger populations, in order to enhance their subjective well-being [24,25,26,27,28]. However, research on how positive psychology affects cognitive functions and brain functioning is still very limited [29].

Positive neuropsychology suggests that positive constructs can influence cognition and brain functioning in various ways [1,2]. Specifically, purpose in life and social support are positively correlated with better cognitive functions across various ethnicities [1,30,31,32,33], sometimes reducing the risk of dementia by up to 30% [30]. Various mechanisms have been suggested. Firstly, regarding neurobiological pathways, it seems that purpose in life can reduce dementia risk through lowering cortisol levels, which are associated with the stress response, and by reducing pro-inflammatory cytokines, which are linked to neuroinflammation and cellular stress [2,33]. Secondly, it has been shown that people with a strong sense of purpose in life are more likely to engage in healthy behaviors, such as physical activity and social engagement, both of which are associated with a reduced risk of dementia. Moreover, eudaimonic wellbeing (associated with purpose in life) may have more protective effects against cognitive decline than hedonic wellbeing (e.g., positive affect) because it promotes goal-oriented behaviors, which require higher cognitive functions such as planning and problem-solving [2,33], as well as resilience, which may buffer the negative effects of stress on cognitive functions [33,34]. For example, mindfulness interventions can enhance cognitive and metacognitive functions and lead to neuroplasticity changes [35,36]. The use of humor can also contribute to the preservation of cognitive functions [37].

Gratitude has been associated with higher cognitive function and structural differences in brain regions, particularly the amygdala and left fusiform gyrus, which are related to emotion and memory, in healthy older people and MCI patients [38]. Humor intervention (improvisation comedy) for healthy older adults showed improvements in mental flexibility, cognitive sharpness and problem-solving skills. Researchers suggested that humor, particularly through active participation in improvisation, can enhance learning, may help maintain mental acuity, and has the potential to delay the onset or progression of dementia [39]. Furthermore, in a sample of adults, it was found that hope—conceptualized as pathways thinking—predicted wisdom, which was defined as integrated dialectical thinking. In turn, wisdom predicted memory performance, while age was found to have a positive influence on both hopeful and wise thinking [40].

Studies on older chronic meditation practitioners (meditation is linked to mindfulness) support that meditation can alter the brain’s function and glucose metabolism [41,42]. Mindfulness is beneficial for reducing cognitive decline because it enhances brain function and structure in areas that are vulnerable to aging by promoting neuroplasticity. This means increasing grey matter density and strengthening functional connectivity in such brain regions. These brain areas include the hippocampus, posteromedial cortex, prefrontal cortex, anterior cingulate cortex, insula, amygdala and default mode network [43].

Different types of meditation can affect the brain in distinct ways. For example, mindfulness meditation seems to lead to increased volume of the hippocampus while loving-kindness meditation, generally, increases the thickness of the posterior and precentral gyri, inferior frontal gyrus, and supramarginal gyri, which are associated with empathy and understanding other’s perspective, but the results are based more on younger and middle-aged people. Perhaps a combination of these two types will lead to better outcomes in more functions [44]. Meditation and mindfulness have also been linked to more regulated functioning within the default mode network in older adults, which may indicate more efficient cognitive functioning [45].

The purpose of the present study is to systematically review the effectiveness of PPIs in patients with SCD, MCI and mild AD dementia in terms of cognitive and brain functioning enhancement. Specifically, the systematic review aims to assess how interventions incorporating positive psychology constructs can influence cognitive functions, brain functioning and, secondarily, well-being and affective symptoms in these populations. Therefore, this systematic review can address the following gap in the literature: although PPIs have been applied to older adults mainly to enhance well-being, their potential to affect cognitive function and brain functioning and thereby delay cognitive decline, particularly in populations with cognitive deficits, is unclear. As a result, the aim of the study is to highlight the potential effects of training in positive constructs on cognitive functions and brain functioning, explore the neuropsychological mechanisms underlying these interventions, and provide insights into the potential benefits of combining training of different constructs (such as mindfulness and specific character strengths) to achieve better outcomes.

## 2. Materials and Methods

### 2.1. Compliance with PRISMA Guidelines

The systematic review was conducted according to the Preferred Reporting Items for Systematic Reviews and Meta-analyses (PRISMA) guidelines [46]. The flow diagram is included. This review was not registered in any database.

### 2.2. Eligibility Criteria

#### 2.2.1. Inclusion Criteria

This review included studies focusing on patients diagnosed with SCD, MCI, or mild AD dementia. Eligible studies implemented PPIs and employed either randomized controlled trial (RCT) designs or pre–post intervention designs with measurable outcomes. A key requirement was that studies assessed at least one of the following: cognitive functions or brain functioning. Finally, only studies written in English were included.

#### 2.2.2. Exclusion Criteria

Studies were excluded if they did not involve an intervention (e.g., correlational studies). Interventions that did not focus primarily on PPIs or those that included positive psychology constructs as part of broader interventions, such as physical activity, cognitive exercise, or mindfulness, were also excluded. Surveys of populations other than those with SCD, MCI, and mild AD dementia were not considered. Finally, research protocols were excluded from the review.

To examine the included studies, they were grouped based on the type of intervention (e.g., mindfulness-based), the population (SCD, MCI, mild AD demented) and the outcome measures (cognitive functions and brain functioning).

### 2.3. Search Strategy

A comprehensive search was conducted between December 2024 and March 2025 across multiple electronic databases, including Scopus, PubMed, ScienceDirect and PsychINFO, to identify relevant studies for the present systematic review. The keywords were designed to capture all studies related to PPIs in patients with SCD, MCI and mild AD dementia and their effects on cognitive functions and brain functioning. Terms such as the following used: “positive psychology intervention”, “MCI”, “cognitive function”, “brain connectivity”, and others were used. A detailed description of the search in each database is provided in Table 1.

The search was limited to studies published between 2015 and 2025, focusing on the subject areas of medicine, neuroscience and psychology, and written in English.

### 2.4. Selection Process

The selection process followed the PRISMA guidelines to ensure a systematic and transparent approach. Two independent reviewers screened all titles and abstracts of the collected articles to evaluate their relevance to the inclusion criteria. They worked independently to minimize bias. If needed, a third reviewer resolved disagreements that couldn’t be resolved through discussion between the two initial researchers. In the second stage, for the remaining records that passed the initial evaluation, full texts were retrieved, and the same two reviewers independently evaluated the full text to avoid bias. Again, any uncertainty about eligibility was resolved through discussion. The reasons for excluding studies in the second stage are documented in detail (Table 2).

Automation tools were used in the selection process to enhance efficiency. Zotero software (version 7.0.13, 64-bit) was used to identify and remove duplicate records. The number of records assessed as ineligible is reported in the PRISMA flow diagram (Figure 1).

All included studies were written in English, so no translation was needed during the selection process.

The final list of included studies was compiled after reviewing full texts. The reviewers cross-checked the included studies to ensure that they met the inclusion criteria. The PRISMA flow diagram represents the entire selection process, including (a) the total number of records identified through database searches, (b) the number of duplicate records identified and removed by Zotero, (c) the number of records screened, assessed and excluded according to eligibility criteria, and (d) the final number of records included.

### 2.5. Data Items

#### 2.5.1. Outcomes

The outcomes for which data were collected were carefully determined to meet the eligibility criteria and align with the goals of the study. The desired outcomes were as follows: first, cognitive functions such as memory, executive functions, and overall cognitive performance decline during cognitive impairment related to dementia. These functions are assessed through neuropsychological tests (e.g., MoCA, MMSE). Second, brain functioning, including measurements of brain activity, structure and connectivity, is assessed through neuroimaging techniques such as fMRI, EEG, fNIRS and MRI.

Data extraction was conducted using a pre-defined data extraction form developed by the authors. The following data elements were extracted from each included study: reference (author, year), design of the study (e.g., RCT, pre–post intervention), population (characteristics of participants, including cognitive status), control group (presence or absence of a control group and type of control group), PPI (type of intervention, duration, frequency), assessment time (time points of assessment), outcome measures (cognitive and brain functioning measures), results (main findings related to cognitive and brain functioning), and the setting where participants were recruited from (community or hospital). Two independent reviewers extracted data, and discrepancies were resolved by consensus.

#### 2.5.2. Other Variables

In addition, other variables were collected to ensure a comprehensive analysis, as outlined below:-Geographic location.-Participant characteristics: mean age and age range, proportion of male and female participants, total number of participants and group sizes, cognitive status of participants (SCD, MCI, mild AD dementia).-Intervention characteristics: specific PPI (e.g., mindfulness intervention), duration (number of weeks and sessions), frequency (e.g., 1 h per week), delivery method (online, in-person, group).-Study characteristics: type of study (RCT, pre–post intervention study), control group (e.g., active control, waitlist, treatment as usual), follow-up period.

No assumptions were made regarding any missing or unclear information.

### 2.6. Synthesis Methods

The process of determining which studies were eligible for inclusion was systematic and aligned with the predefined criteria outlined above. Due to the limited number of studies, a quantitative meta-analysis was not possible. Therefore, we conducted a qualitative synthesis of the data, summarizing the key findings and identifying common themes across studies. The steps are outlined below:Tabulating study characteristics: A table (Table 3) was created to summarize the key points of each study. Specifically, this table includes information about geographic location, design of the study (e.g., RCT, pre–post intervention study), population (e.g., SCD, MCI, mild AD demented), control group (no/active control group, waitlist, treatment as usual), PPI (e.g., mindfulness intervention), the setting where participants were recruited from (community or hospital).Tabulating the effectiveness of PPI interventions: A table (Table 4) was created to summarize the effectiveness of each study. This table contains assessment time (pre, post, follow-up), outcome measures and results.Comparing study characteristics against planned groups: studies were grouped based on type of PPI.Reviewer 1 and Reviewer 2 worked independently to group the results, and then they compared and discussed the findings.

## 3. Results

### 3.1. Study Selection

The PRISMA flow diagram in Figure 1 above presents the study selection process. The total number of records identified through the database search was 887. Duplicates were removed using the software Zotero, resulting in a total of 387 unique records. During the title and abstract screening phase, 311 records were excluded. These 311 articles were excluded during the screening phase because, based on their titles and abstracts, the reviewers determined that they did not meet the inclusion criteria outlined above. The reviewers conducted this screening process manually without the use of any automated tool. The total number of full-text articles assessed for eligibility was 76, but 64 were excluded because they did not meet the inclusion criteria. The specific reason for the exclusion of each article at this stage is detailed in Table 2.

Finally, the systematic review included 12 studies (journal articles) that investigated the effects of PPIs on cognitive functions (e.g., memory and executive functions) and brain functioning (e.g., brain volume, brain connectivity and inflammatory biomarkers).

Analyzing reasons for exclusion:

No PPI: include studies with interventions other than PPI, studies that incorporate constructs of positive psychology along with other elements, or studies with no intervention at all.

Other population: studies that do not include people with SCD, MCI and mild AD dementia.

Protocols: study protocols.

Other outcomes measured: studies in which outcomes other than cognitive functions and brain functioning are measured.

### 3.2. Excluded Studies

During the full-text review, 64 studies were excluded. For example, the study of Schlosser et al. [59] was excluded because they did not measure cognitive and brain outcomes, and the study of Namias et al. [60] was also excluded because it included a different population. A complete list of excluded studies and reasons for exclusion is provided in Table 2.

### 3.3. Study Characteristics

Study characteristics are summarized in Table 3. The studies were conducted in various countries, such as Spain, Singapore, and Australia, and involved different types of interventions, outcome measures and populations. Eleven of them [47,48,49,50,51,52,53,54,56,57,58] were RCTs, and one used a pre–post-intervention design [55]. All interventions included different forms of mindfulness-based interventions (e.g., Mindfulness-Based Training, Mindful Awareness Practice, and Caring Mindfulness-Based Approach for Seniors), and some also incorporated psychoeducation and various healthcare programs. No interventions specifically targeting character strengths were identified in these populations. Assessment time varied across different studies, but most studies included baseline, pre–post intervention assessment, and some of them included follow-up assessment at 3, 6, or 9 months too. The effectiveness of each intervention is summarized in Table 4.

### 3.4. Participants

The total number of studies included in the systematic review, according to eligibility criteria, is 12. The total number of participants included across these 12 studies is 669. In two studies [50,51], the participants were the same, so they were counted only once. Nine studies [49,50,51,52,53,54,55,56,58] included MCI participants, and more specifically, Larouche et al. [53] used amnestic MCI (aMCI) participants. Two studies [47,48] examined SCD participants, while Smart and Segalowitz [48] included health controls, too. One study [57] included probable AD dementia participants. Participants were adults aged 56 years old and older. The studies were conducted between 2016 and 2022.

### 3.5. Interventions

Regarding interventions, they were conducted in person and in groups. Eleven out of twelve studies [47,48,49,50,51,52,53,54,55,56,58] included weekly sessions. Five of them [47,51,53,54,57] were completed in 8 weeks, six of them [49,50,51,52,53,58] in 12 weeks, among them five [50,51,53,54,58] included a monthly meeting for the next 6 months. The remaining study [57] included 3 weekly sessions for 3 years. Participants were expected to practice the content of the interventions at home.

### 3.6. Results of Synthesis

The following units present results from studies regarding the effects of interventions on cognitive functions and brain functioning. Results are separated into units according to the type of Mindfulness intervention, as named in the studies. All the interventions share a common background, namely, a focus on mindfulness. However, they are categorized according to the type of intervention. Even interventions with a similar structure have been classified as separate interventions, depending on whether they are based on the same theory literature and whether they differ in duration. This is because the available data about these interventions are very limited, and a clear separation can better help to understand which aspects of the interventions influence their effectiveness in cognitive functions and brain functioning. The main findings on the effectiveness of all interventions regarding cognitive functions and brain functionality are presented in Table 5.

#### 3.6.1. Mindful Awareness Practice (MAP)

Mindfulness Awareness Practice (MAP) [50,51,58], sometimes referred to as Mindfulness Awareness Program [52,53,54] or MIND [52], was inspired by McBee’s [61] mindfulness techniques (MBEC) adapted to the unique needs of older adults. Sessions were guided by an experienced instructor, who trained participants in mindfulness techniques such as mindfulness of the senses, body scan, walking meditation practice, movement nature meant to practice and visuomotor limb tasks. Participants were required to practice these techniques daily at home and record their practice. The intervention consisted of 12 weekly sessions followed by monthly booster sessions for six months.

MAP has been examined in 6 studies in MCI patients. MAP appears to be promising in influencing biological pathways, as evidenced by enrichment in pathways related to immune response, inflammation, and cellular stress, with 68 genes differentially expressed, which are implicated in cognitive decline and dementia [58]. In addition, MAP has been shown to reduce inflammatory biomarkers [50], but these biological changes do not consistently translate into long-term clinical improvements in general cognitive functions and affective symptoms [51]. Short-term benefits have been observed in specific cognitive domains such as recognition memory [52], working memory span, and divided attention [54]. Neuroplasticity changes have been observed, too, with temporary increases in cortical thickness in the left inferior temporal gyrus at 3 months. However, these changes in the inferior temporal gyrus are not sustained at 9 months of assessment, where participants exhibited increased cortical thickness in the right frontal pole and decreased cortical thickness in the left anterior cingulate cortex [54]. Klainin–Yobas et al. [53] suggested that MAP may improve affective symptoms, but the Health Education Program (HEP) often yields superior results. Consequently, MAP may have potential short-term beneficial effects in some specific domains of cognition and biological processes, but its long-term effects on general cognitive functions and affect remain unclear, limited and inconsistent.

#### 3.6.2. Mindfulness-Based Training (MBT)

MBT [49] consisted of 8 weekly sessions. Each session included an introduction and practice of mindfulness exercises. Sessions also included discussion of participants’ experiences, how mindfulness is correlated to cognitive abilities and assignments for home. They were required to practice mindfulness techniques for at least 15 min per day at home and record their practice in diaries. Audio guides for mindfulness techniques were provided.

The 8-week MBT, as described by Doshi et al. [49], showed significant improvements in global cognition as measured by MMSE and delayed memory as measured by RBANS in MCI patients. However, since the study included both active (Cognitive Rehabilitation Therapy—CRT) and passive (Treatment as Usual—TAU) control groups, MBT did not demonstrate greater improvement in cognitive functioning than spontaneous reversion rates. Therefore, MBT’s superiority compared to natural, spontaneous cognitive improvement has not been well documented.

#### 3.6.3. Mindfulness Training (MT)

MT [48] was an 8-week mindfulness program based on Kabat-Zinn’s [62] mindfulness-based stress reduction, but specifically tailored for older adults. Half of the program consisted of training in focused-attention mindfulness techniques, the other half in open-monitoring mindfulness, and two practices related to emotion regulation (emotional weather and loving-kindness). Participants were also introduced to the concept of cognitive reserve and openness to new activities. They were given audio guides and workbooks and required to practice at home daily.

Smart and Segalowitz [48] found that an 8-week MT significantly increased Error-Related Negativity (ERN) amplitude, compared to the Psychoeducation (PE) group, which led to improved attentiveness to errors and monitoring performance. On the other hand, participants in the PE group showed increased Error Positivity (Pe) amplitude, leading to better emotional reactivity to errors. Participants were people with SCD and healthy controls.

#### 3.6.4. Mindfulness Training Program (MTP)

MTP [55] was an 8-week mindfulness program adapted to MCI patients and guided by experienced facilitators. It consisted of both formal (e.g., body scan, breath meditation, maintaining attention with no judgment and loving-kindness meditation) and informal (mindfulness in everyday life) training. Participants were given audio recordings in order to practice daily at home.

MTP intervention in MCI patients conducted by Wong et al. [55] led to significant improvements in cognitive functions even a year after the end of the intervention. They also found increased trait mindfulness and daily living functioning.

#### 3.6.5. Mindfulness-Based Intervention (MBI)

MBI [56] was an 8-week program based on Kabat-Zinn’s mindfulness stress reduction [62] and Segal et al.’s [63] mindfulness-based therapy, along with tools from other sources, adapted for aMCI patients. Participants were trained in mindfulness techniques and were required to practice both formal (mindfulness techniques) and informal (mindfulness in everyday life) techniques at home. They were also required to record their practice.

Both 8-week MBI and Psychoeducation-Based Intervention (PBI) had similar effects on significantly reducing symptoms of anxiety and depression and improving aging-related quality of life, but not episodic memory performance, in older adults with aMCI [56].

#### 3.6.6. Mindfulness-Based Alzheimer’s Stimulation (MBAS)

MBAS [57] was inspired by the MBSR program, MBEC, Kirtan Kriya technique, chair-yoga exercises, and sensorial integration theory. Formal training, meaning mindfulness techniques, took place during sessions with patients and their caregivers. A typical session consisted of orientation exercises, chair yoga, attention to breathing, body scan and Kirtan Kriya techniques. Informal training, meaning the use of mindfulness in everyday activities such as walking, took place at home, where caregivers dedicated 10 min every day to practicing with patients the techniques learned during the sessions. The duration of the program was 2 years (288 sessions) with weekly group sessions.

Quintana-Hernández et al. [57] examined the effectiveness of MBAS in older patients with probable AD dementia compared to Cognitive Stimulation Therapy (CST), Progressive Muscle Relaxation (PMR), and a pharmacological treatment-only control group (Donepezil). All patients in all groups were treated with Donepezil. The most effective interventions in maintaining cognitive capacities were MBAS and CST. MBAS demonstrated the most consistent and stable effects in all cognitive domains measured (memory, attention, language, praxis) over a two-year period.

#### 3.6.7. Caring Mindfulness-Based Approach for Seniors (CMBAS)

CMBAS [47] was based on the MBSR with weekly meditation practice combined with compassion meditation practices and psychoeducation. In addition, participants were required to practice daily at home for 1 h, using both formal (e.g., guided meditation) and informal (mindful eating) techniques. The total duration was 8 weekly group sessions and a half-day of meditation practice during the sixth week.

Whitfield et al. [47] assessed the effectiveness of CMBAS in comparison to a Health Self-Management Program (HSMP) in SCD patients. They found that both interventions resulted in significant improvements in global cognition, as measured by the PACC5Abridged, with no significant difference between the two groups at week 24. There was weak evidence of improvement in the attention composite but no measurable improvements in the executive function composite in either group.

### 3.7. Quality Appraisal

Quality appraisal of the included studies was conducted using the Mixed Methods Appraisal Tool (MMAT), version 2018. The MMAT is a critical appraisal tool designed to assess the methodological quality of qualitative, quantitative, and mixed methods studies within systematic reviews. It provides a set of criteria tailored to different study designs, enabling a comprehensive and consistent evaluation process [64]. The procedure followed for the quality appraisal is presented in Table 6.

Overall, the included studies were of moderate quality. Among the 11 randomized controlled trials, randomization was appropriately performed in all studies (*n* = 11; 100%), and baseline comparability was achieved in most (*n* = 10; 91%). However, complete outcome data were available in only one study (*n* = 1; 9%), with the majority experiencing attrition or missing data. Blinding of outcome assessors was reported in just over half of the RCTs (*n* = 6; 55%), while adherence to the assigned intervention was satisfactory in most (*n* = 9; 82%). The single quantitative descriptive study [55] met the criteria for appropriate measurements and analysis but did not achieve a representative sample or low risk of nonresponse bias. These findings highlight that, while randomization and intervention fidelity were generally well addressed, future research should prioritize minimizing attrition, improving blinding, and enhancing sample representativeness to strengthen the evidence base for mindfulness interventions in older adults.

## 4. Discussion

The purpose of the present systematic review was to collect, examine and synthesize the findings of studies that applied PPIs to individuals with SCD, MCI and mild AD dementia and examined their effects on cognitive functions and brain functioning. In this way, the present study aimed to examine the potential of PPIs as non-pharmacological interventions in cognitive decline related to dementia. Positive neuropsychology is an emerging scientific field that studies the effects of positive psychology on brain functioning and cognitive functions [1,2]. Although research, until now, is extremely limited, correlations have been found demonstrating that such interventions could contribute to this goal [1,30,31,32,33]. Such interventions could be based on CS [37,38] and mindfulness [36]. Our aim was to investigate all studies that used different constructs of positive psychology to enhance cognitive functions and brain functioning in individuals with cognitive decline related to dementia. However, our systematic review focuses on mindfulness, as it appears that, so far, only such studies have been conducted. This highlights a significant gap in the literature.

It is crucial to consider the methodological heterogeneity across the included studies. These studies differed significantly in terms of intervention protocols (e.g., duration, intensity, specific mindfulness techniques used) [52,55,57], outcome measures (e.g., different cognitive tests, various neuroimaging modalities) [48,52,54], sample characteristics (e.g., varying degrees of cognitive impairment, different inclusion/exclusion criteria) [47,53,57], and control groups (e.g., active vs. passive controls, pharmacological controls, or no control group) [49,55,57]. This heterogeneity makes it challenging to directly compare and synthesize the results and limits the generalizability of the findings.

Mindfulness can lead to increased cognitive functions and brain functioning, highlighting its potential as a low-cost, non-pharmacological intervention for dementia. Participants in MAP showed preserved or improved temporal global efficiency, meaning that mindfulness can help the brain’s ability to transmit information more efficiently and quickly. In addition, mindfulness was found to enhance dynamic brain connectivity. These changes are likely the result of increased grey matter induced by mindfulness practice [52]. Brain areas such as the insula, anterior cingulate gyrus and superior temporal gyrus demonstrated improved localized temporal nodal efficiency. These regions are associated with self-regulation, interoceptive awareness, and auditory processing. Enhanced connectivity in these areas suggests that mindfulness may strengthen the functional integration of these regions [52,54].

Furthermore, MAP increased cortical thickness in the right frontal lobe and inferior temporal gyrus, brain areas associated with cognitive control and memory. At the same time, decreased cortical thickness in the left anterior cingulate gyrus was linked to reduced emotional reactivity and better emotional regulation [54]. The results of MAP are likely due to the repeated engagement and disengagement of attention during mindfulness practice, which may lead to improvements in working memory span and divided attention [54]. Furthermore, MAP improved inflammatory biomarkers such as CRP, IL-6 and IL-1β, showing its potential to target biological pathways associated with dementia [50]. Changes in gene expression were also observed, meaning enrichment in pathways related to neuroactive ligand-receptor interactions and stress-related pathways. This could happen due to the upregulation and downregulation of gene expression during mindfulness practice [58].

Regarding cognitive functions, MAP leads to better memory recognition. The process of recognition is not as cognitively demanding as the process of recall, making recognition more sensitive to improvement. Group-based mindfulness practice may also contribute to these improvements because of experiential learning and socio-emotional sharing [52].

While MAP showed short-term effectiveness in enhancing neuroplasticity and specific cognitive functions, its long-term effectiveness in general cognitive function remains unclear, as Ng et al. [51] reported in a 5-year follow-up. These outcomes may result from the difficulty of transferring these practices to other domains (e.g., in everyday life) and other challenges that may come up in the long term.

Quintana-Hernández et al. [57] suggested that the combination of MBAS with donepezil was the most effective non-pharmacological intervention for mild-to-moderate-stage AD across all cognitive functions measured (sense of direction, language, memory, perception, attention, and praxia). This can be explained by the association of mindfulness practice with increased cortical thickness [65], changes in the default mode network [66] and activation in brain areas related to attention, awareness and emotion regulation [67,68]. Also, mindfulness has been linked to decreased risk factors for dementia, such as stress and hypertension [69,70]. However, it is important to note that this study used a specific combination of MBAS and donepezil, making it difficult to isolate the independent effects of MBAS.

MT led to increased ERN amplitude and, as a result, enhanced attentiveness to errors. According to Smart and Segalowitz [48], mindfulness increases self-regulation by teaching participants to monitor their response to errors without judgment or impulsivity, leading to specific neural changes. MBT led to better-delayed memory [49] and improved global cognitive functioning, although not significantly greater than the control group, same as CMBAS [47,49]. This outcome is probably the result of diagnostic instability. Doshi et al. [49] suggest that they may have needed to focus on other factors to measure mindfulness’ effectiveness, such as the decrease of risk factors (e.g., depression). They also suggest that mindfulness practice may be more efficient in highly educated people because of its requirements. CMΒAS intervention improved attention but not significantly more than the active control group. This finding is also supported by other studies [47,48]. The lack of significant improvements in executive functions can be explained by the absence of working memory training, which is primarily correlated to such improvements. As long as the results are closely similar between mindfulness and active control groups, this can be attributed to common mechanisms between them, such as social interaction and behavioral activation [47,56]. A longer duration of MTP intervention can enhance cognitive functions and increase participants’ everyday functioning [55]. Similarly, others have suggested that shorter-duration programs, such as 8-week programs, may be inefficient in enhancing cognitive functions and memory due to the limited training provided to the participants [53,56].

Mindfulness mechanisms can also be explained by Monitor and Acceptance Theory (MAT) [71], which describes two key pathways through which mindfulness acts: acceptance and monitoring (observation). Larouche et al. [56] found that acceptance helped mitigate judgmental attitudes and hyperreactivity, leading to a reduction in depressive and anxiety symptoms, partly through decreased ruminations. However, monitoring did not positively affect participants in any way.

Therefore, the findings from the present systematic review highlight the potential of mindfulness interventions to improve specific cognitive functions in people with SCD, MCI and mild AD dementia. A key concern is whether these interventions can be more effective than other, more traditional interventions and in what specific form [49]. The evidence suggests that both duration and continuous practice are crucial for consolidating results and demonstrating effectiveness. For example, Whitfield et al. [47] showed that improvements in global cognition were significant in week 24 and not in week 8, although in other studies, improvements were observed in 3 months but diminished in 9 months. This may be due to the lack of consistency in practice [51]. Wong et al. [55] suggested that a longer duration of mindfulness intervention led to improvements in cognitive functions and everyday activities even at a one-year follow-up.

Beyond the specific populations, research to date has shown that constructs of positive psychology, such as meaning in life and eudaimonic well-being, can enhance cognitive functions and brain functioning. In terms of duration, as shown by the review and based on the existing literature, longer interventions are more effective and produce more durable results over time, as learning requires time and practice [19,72,73]. Still, multidomain cognitive interventions appear to work better, as they activate more areas of the brain [74]. Similarly, multidomain PPIs, which use a range of positive constructs, are probably more effective [28,75] and even more effective over longer periods [76]. As mindfulness interventions have been shown to be potentially effective in enhancing cognitive functions and brain functioning in people with SCD, MCI and mild AD dementia, it is possible that the same could be true for CS. This is because mindfulness and CS are interrelated, and the effective use of mindfulness requires CS and vice versa [23]. Some mindfulness interventions already include character strengths such as loving-kindness [48,55] and curiosity [49], although not explicitly. In this way, their combined training could prove to be the best choice.

### 4.1. Implication for Future Research

Therefore, the data so far, although limited, are encouraging. More studies are needed that examine the effectiveness of a range of these interventions in people with preclinical cognitive deficits, focusing on cognitive and brain functioning rather than simply correlating these outcomes with positive psychology data. In the future, research should incorporate more positive psychology constructs and extend the duration of studies to investigate in depth the mechanisms and effects of these interventions. For example, longer (e.g., 6 months with weekly sessions) and multidomain (e.g., different character strengths and mindfulness) interventions can be applied to older adults in preclinical stages of dementia to examine their effectiveness in cognitive functions and brain functioning. Finally, these interventions could also be delivered online to be more accessible to everyone, regardless of their location.

### 4.2. Limitations

The limited amount of research in the domain of interest makes the present systematic review a primary effort to motivate further research. The heterogeneity of study designs and outcome measures did not allow for definitive conclusions. Also, the lack of standardized protocols and the use of diverse control groups made it challenging for the results to be comparable across studies. Still, the review focuses solely on mindfulness interventions and does not include other constructs of positive psychology.

Despite the fact that the results are encouraging, only 12 studies were identified, and therefore, the available data are very limited and difficult to generalize. Additionally, the fact that such interventions have only been implemented in developed countries further reduces their generalizability to countries with different socioeconomic levels and cultures. In addition, the implementation of mindfulness interventions presents its own challenges, such as the need for considerable time for learning, commitment to systematic practice, and, of course, the availability of trained instructors.

It is also important to acknowledge that many of the included studies had small sample sizes and lacked blinding, which may introduce bias and limit the reliability of the findings. The use of different cognitive assessments and biomarkers across studies also complicates the interpretation and comparison of results.

Finally, the interpretation of some findings is complicated by the fact that many studies tested multiple variables and corrections for multiple comparisons were not consistently applied. This raises the possibility that some reported significant results may be due to chance and should be interpreted with caution.

## 5. Conclusions

This review was an attempt to evaluate the effectiveness of positive psychology interventions as non-pharmacological interventions for cognitive decline related to dementia. The available data on the effectiveness of these interventions is very limited, focusing primarily on mindfulness, a single positive psychology construct. Nevertheless, the data are quite encouraging, suggesting a positive impact on patients’ cognitive functions and brain functioning. Further research is needed to systematically examine more experimental interventions.

## Figures and Tables

**Figure 1 brainsci-15-00580-f001:**
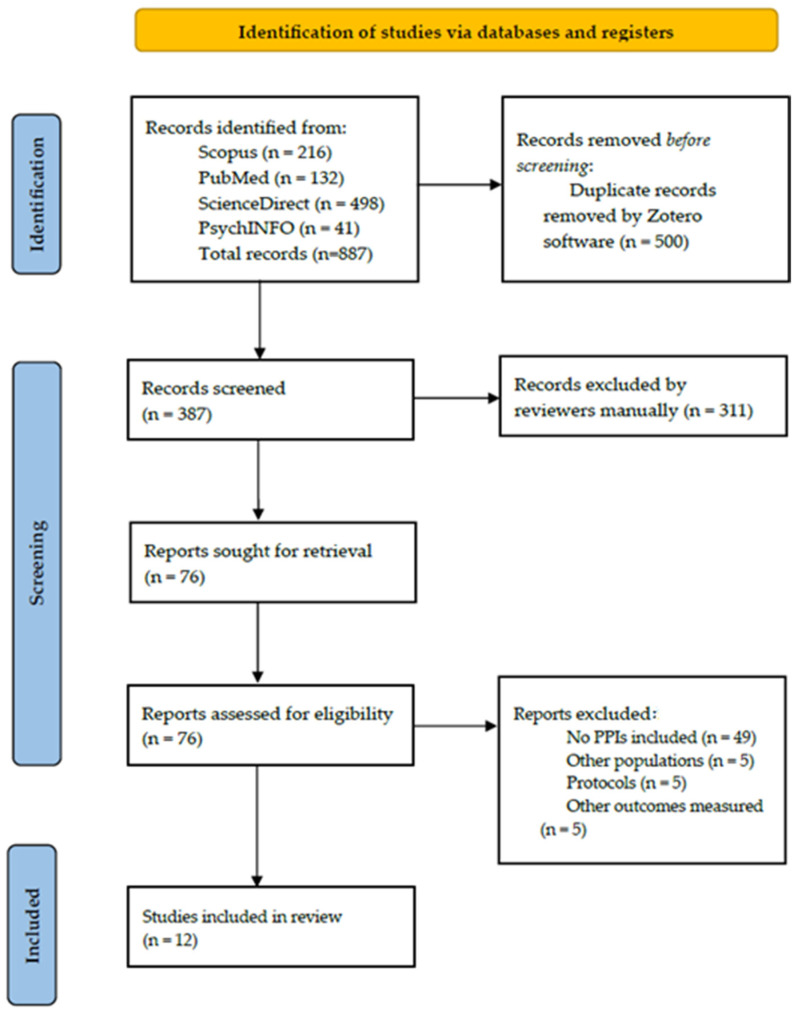
PRISMA flow diagram of Study Selection Process.

**Table 1 brainsci-15-00580-t001:** Search Terms.

Data Base	Search Term
Scopus	(TITLE-ABS-KEY (“positive psychology” OR “positive neuropsychology” OR “positive neuroscience” OR “positive psychology intervention” OR “psychological intervention” OR “mindfulness” OR “gratitude intervention” OR “meaning in life” OR “purpose in life” OR “well-being intervention” OR “hedonic well-being” OR “eudaimonic well-being” OR “character strengths” OR “virtues” OR “signature strengths” OR “strengths-based intervention”)) AND (TITLE-ABS-KEY (“mild cognitive impairment” OR “MCI” OR “subjective cognitive decline” OR “Alzheimer’s disease” OR “early-stage dementia” OR “mild dementia”)) AND (TITLE-ABS-KEY (“cognitive function” OR “cognitive decline” OR “memory” OR “executive function” OR “cognitive reserve” OR “brain function” OR “brain connectivity” OR “functional connectivity” OR “neural plasticity” OR “fMRI” OR “EEG” OR “fNIRS” OR “cortex” OR “prefrontal cortex” OR “anterior cingulate cortex” OR “temporal cortex” OR “neuroplasticity”)) AND PUBYEAR > 2014 AND PUBYEAR < 2026 AND (LIMIT-TO (DOCTYPE, “ar”) OR LIMIT-TO (DOCTYPE, “re”) OR LIMIT-TO (DOCTYPE, “ch”)) AND (LIMIT-TO (SUBJAREA, “MEDI”) OR LIMIT-TO (SUBJAREA, “NEUR”) OR LIMIT-TO (SUBJAREA, “PSYC”)) AND (LIMIT-TO (LANGUAGE, “English”)) Limited to: Title-Abstract-Keywords Areas of neuroscience, medicine and dentistry and psychology Publication Date: 2015–2025
PubMed	(“positive psychology” OR “positive neuropsychology” OR “positive neuroscience” OR “positive psychology intervention” OR “positive psychological intervention” OR “mindfulness” OR “gratitude intervention” OR “meaning in life” OR “purpose in life” OR “well-being intervention” OR “hedonic well-being” OR “eudaimonic well-being” OR “character strengths” OR “virtues” OR “signature strengths” OR “strengths-based intervention”) AND (“mild cognitive impairment” OR “MCI” OR “subjective cognitive decline” OR “Alzheimer’s disease” OR “early-stage dementia” OR “mild dementia”) AND (“cognitive function” OR “cognitive decline” OR “memory” OR “executive function” OR “cognitive reserve” OR “brain function” OR “brain connectivity” OR “functional connectivity” OR “neural plasticity” OR “fMRI” OR “EEG” OR “fNIRS” OR “cortex” OR “prefrontal cortex” OR “anterior cingulate cortex” OR “temporal cortex” OR “neuroplasticity”) Areas of neuroscience, medicine and dentistry and psychology Publication Date: 2015–2025
ScienceDirect	(“positive psychology” OR “character strengths” OR “mindfulness”) AND (“mild cognitive impairment” OR “subjective cognitive decline” OR “Alzheimer’s disease”) AND (“cognitive function” OR “brain function” OR “cortex”) Publication Date: 2015–2025
PsychINFO	1. “positive psychology” OR “positive neuropsychology” OR “positive neuroscience” OR “positive psychology intervention” OR “positive psychological intervention” OR “mindfulness” OR “gratitude intervention” OR “meaning in life” OR “purpose in life” OR “well-being intervention” OR “hedonic well-being” OR “eudaimonic well-being” OR “character strengths” OR “virtues” OR “signature strengths” OR “strengths-based intervention”
	2. “mild cognitive impairment” OR “MCI” OR “subjective cognitive decline” OR “Alzheimer’s disease” OR “early-stage dementia” OR “mild dementia”)
	3. “cognitive function” OR “cognitive decline” OR “memory” OR “executive function” OR “cognitive reserve” OR “brain function” OR “brain connectivity” OR “functional connectivity” OR “neural plasticity” OR “fMRI” OR “EEG” OR “fNIRS” OR “cortex” OR “prefrontal cortex” OR “anterior cingulate cortex” OR “temporal cortex” OR “neuroplasticity” Limited to: Abstract Publication Date: 2015–2025

**Table 2 brainsci-15-00580-t002:** List of excluded articles.

Item Type	Publication Year	Author	DOI	Exclusion Reason
journalArticle	2023	Willroth et al.	10.1177/09567976221119828	No PPI *
journalArticle	2024	Namias et al.	10.1080/20445911.2024.2381872	Other population **
journalArticle	2024	Ridder et al.	10.1080/08098131.2024.2430789	No PPI
journalArticle	2022	Ismail et al.	10.1177/00914150211024185	No PPI
journalArticle	2022	Ng et al.	10.1111/appy.12518	No PPI
journalArticle	2024	D’elia et al.	10.1002/dad2.12558	Other outcomes measured ***
bookSection	2024	Khalsa et al.		No PPI
journalArticle	2023	Nicosia et al.	10.1177/27536130231202989	No PPI
journalArticle	2023	Sandison et al.	10.3233/JAD-230004	No PPI
journalArticle	2025	Šumec et al.	10.1007/s12671-024-02500-9	No PPI
journalArticle	2021	Galvin et al.	10.3233/JAD-215077	No PPI
journalArticle	2024	Yang et al.	10.1186/s12877-024-05090-2	Protocol ****
journalArticle	2023	Wu et al.	10.1186/s12883-023-03390-5	No PPI
journalArticle	2022	Dewitte et al.	10.1007/s10433-022-00689-z	No PPI
journalArticle	2024	O’Shea et al.	10.3233/JAD-231346	No PPI
journalArticle	2023	Strikwerda-Brown et al.	10.1016/j.bpsgos.2022.01.001	No PPI
journalArticle	2025	Lewis et al.	10.1080/13825585.2024.2373846	No PPI
journalArticle	2024	Shim et al.	10.1080/13607863.2024.2364754	Other outcomes measured
journalArticle	2023	Pluim et al.	10.1177/07334648221139479	No PPI
journalArticle	2015	Khalsa et al.	10.3233/JAD-142766	No PPI
journalArticle	2022	10.1016/j.cct.2025.107811	10.1093/gerona/glac093	No PPI
journalArticle	2022	Jopowicz et al.	10.3390/brainsci12030345	No PPI
journalArticle	2022	Stewart et al.	10.1177/07334648221095514	No PPI
journalArticle	2023	Quintana-Hernández et al.	10.3233/JAD-220889	Other outcomes measured
journalArticle	2022	Mace et al.	10.1007/s10880-022-09843-2	No PPI
journalArticle	2023	Abellaneda-Pérez et al.	10.1186/s13195-023-01198-6	No PPI
journalArticle	2022	Lenze et al.	10.1001/jama.2022.21680	No PPI
journalArticle	2016	Acevedo et al.	10.1007/s40473-016-0098-x	No PPI
journalArticle	2018	Dos Santos et al.	10.3389/fpsyg.2018.00371	No PPI
journalArticle	2021	Lutz et al.	10.1016/j.arr.2021.101495	No PPI
journalArticle	2022	Vespa et al.	10.1007/s40520-021-01907-x	Protocol
journalArticle	2021	Marchant et al.	10.1159/000515669	Other outcomes measured
journalArticle	2017	Nicholson et al.	10.1177/0091415017709789	No PPI
journalArticle	2018	Ismail et al.	10.3233/JAD-180075	No PPI
journalArticle	2020	Schlosser et al.	10.1186/s12888-020-02884-7	No PPI
journalArticle	2021	Khalsa et al.	10.3233/JAD-201433	No PPI
journalArticle	2020	McDonough et al.	10.1037/neu0000606	No PPI
journalArticle	2020	Chan et al.	10.1080/14737175.2020.1810571	No PPI
journalArticle	2018	Lardone et al.	10.1155/2018/5340717	No PPI
journalArticle	2021	Reynolds et al.	10.1016/j.amjmed.2020.10.041	No PPI
journalArticle	2019	Nakanishi et al.	10.3233/JAD-190590	No PPI
journalArticle	2022	MacAulay et al.	10.1080/13607863.2021.1998352	No PPI
journalArticle	2018	Chételat et al.	10.1186/s13195-018-0388-5	No PPI
journalArticle	2018	Zahodne et al.	10.1017/S1355617717000935	No PPI
journalArticle	2021	Sevinc et al.	10.3389/fnagi.2021.702796	Other population
journalArticle	2018	Marchant et al.	10.1016/j.trci.2018.10.010	Protocol
bookSection	2017	Crescentini et al.		No PPI
journalArticle	2016	Trivedi et al.	10.1177/0891988715598235	No PPI
journalArticle	2023	Kawada et al.	10.1007/s40520-023-02377-z	No PPI
journalArticle	2023	Schlosser et al.	10.1371/journal.pone.0295175	Other outcomes measured
journalArticle	2022	Wagner et al.	10.1093/geroni/igac019	No PPI
journalArticle	2017	Cook Maher et al.	10.1371/journal.pone.0186413	No PPI
journalArticle	2015	Yu et al.	10.1161/STROKEAHA.114.008010	No PPI
journalArticle	2025	Friedman et al.	10.1093/geronb/gbaf021	No PPI
journalArticle	2017	Morse et al.	10.1016/j.archger.2017.10.013	Other population
journalArticle	2024	Choukas et al.	10.1016/j.archger.2023.105290	No PPI
journalArticle	2025	Huang et al.	10.1016/j.gerinurse.2025.01.011	No PPI
journalArticle	2023	Liu et al.	10.1016/j.jagp.2022.10.006	No PPI
journalArticle	2025	Chao et al.	10.1016/j.cct.2025.107811	Protocol
journalArticle	2023	Kim et al.	10.1016/j.pmedr.2023.102165	No PPI
journalArticle	2021	Yang et al.	10.1016/j.pmedr.2021.101490	Other population
journalArticle	2018	Poisnel et al.	10.1016/j.trci.2018.10.011	Protocol
journalArticle	2024	Kaliman et al.	10.1016/j.bpsgos.2024.100398	Other population
bookChapter	2019	Deepak	10.1016/bs.pbr.2018.10.030	No PPI

* No PPI: include studies with interventions other than PPI, studies that incorporate constructs of positive psychology along with other elements, or studies with no intervention at all; ** Other population: studies that do not include people with SCD, MCI and mild AD dementia; *** Other outcomes measured: studies in which outcomes other than cognitive functions and brain functioning are measured; **** Protocol: study protocols.

**Table 3 brainsci-15-00580-t003:** Characteristics of included studies.

S/N	Reference	Design	Population	Control Group	PPI	Recruited from
1	Whitfield et al., 2022 (Europe) [47]	Randomized controlled trial (RCT)	147 older adults with SCD (Mean age: 72.7 years, SD: 6.9) from memory clinics: - 73 (47 females) in the CMBAS group and - 74 (48 females) in the HSMP group.	Active control group	- Caring Mindfulness-Based Approach for Seniors (CMBAS): A mindfulness-based intervention followed the general format of a mindfulness-based stress reduction program. - Health Self-Management Program (HSMP): An active comparator multidomain intervention. - Both were group-based, 8-week programs with similar formats (weekly 2-h sessions and a half-day session in week six), conducted in person.	Memory clinics
2	Smart and Segalowitz, 2017 (Canada) [48]	Randomized controlled trial (RCT)	- 32 older adults aged 65–80 years. - MT group: 16 participants (10 females; 6 SCD, 10 Health Control). - AC group: 16 participants (11 females; 7 SCD, 9 Health Control).	Active control group	- MT: Mindfulness Training (8-week, in-person, weekly group sessions with home practice). - AC: Psychoeducation (5-week, in-person, weekly group sessions with home practice)	Community
3	Doshi et al., 2021 (Singapore) [49]	Randomized controlled trial (RCT)	- 76 older adults with MCI (Mean age: 67 years, SD: 5.3). - MBT: 32 participants (16 males, 16 females). - CRT: 27 participants (10 males, 17 females). - TAU: 17 participants (8 males, 9 females). - The randomization ratio was adjusted to 2:2:1 due to recruitment challenges.	Both active and passive control groups	- MBT: Mindfulness-Based Training (an 8-week program focused on cognitive benefits). - CRT: Active control group with cognitive rehabilitation training. - TAU: The passive control group is receiving standard care treatment as usual. - Both MBT and CRT included 8 weekly 1.5–2-h sessions with home practice, conducted in person.	Both a hospital memory clinic and the community
4	Ng et al., 2020 (Singapore) [50]	Randomized controlled trial (RCT)	- 55 older adults with MCI (Mean age: 71.28 years, SD = 6.00). - MAP: 28 participants (20 females). - HEP (Control): 27 participants (21 females). - MCI subtypes: 21 aMCI (38.2%) and 34 naMCI (61.8%).	Active control group	- MAP: Mindful Awareness Practice. - HEP: Health Education Program (Control). - Both included weekly 1-h in-person sessions for 12 weeks, followed by six monthly booster sessions from months 3 to 9.	Community-based research center
5	Ng et al., 2022 (Singapore) [51]	Randomized controlled trial (RCT)	- 55 community-dwelling older adults aged 60–85 years with MCI (Mean age: 71.3 years, SD = 6). - MAP: 28 participants (20 females). - HEP: 27 participants (21 females).	Active control group	- MAP: Mindful Awareness Practice. - HEP: Health Education Program (Active Control). - Both included weekly in-person sessions for 3 months, followed by monthly booster sessions for 6 months.	Community-based research center
6	Fam et al., 2020 (Singapore) [52]	Randomized controlled trial (RCT)	- 36 older adults (aged 60–85 years) with MCI. - MIND: 19 participants (12 females, Mean age = 72.58 years, SD = 5.24). - CTRL: 17 participants (14 females, Mean age = 70.71 years, SD = 6.00).	Active control group	- MIND: Mindfulness Awareness Program with weekly 40-min in-person sessions for 12 weeks, including daily home practice and record-keeping. - CTRL: Weekly 40-min in-person health education talks for 12 weeks on topics like diet, sleep, exercise, and home safety, with daily activity record-keeping.	Community
7	Klainin-Yobas et al., 2019 (Singapore) [53]	Randomized controlled trial (RCT)	- 55 older adults (aged 60–85 years) with MCI. - Mean age: 71.35 years (SD = 5.76). - MAP: 28 participants (20 females). - HEP: 27 participants (21 females).	Active control group	- Mindfulness Awareness Program (MAP): Weekly 40-min mindfulness sessions for 3 months, followed by monthly sessions for 6 months, with home practice encouraged. - Health Education Program (HEP): Weekly 40-min health education sessions for 3 months, followed by monthly sessions for 6 months, with application of topics to daily routines encouraged. - Both were conducted in person.	Community
8	Yu et al., 2021 (Singapore) [54]	Randomized controlled trial (RCT)	- 54 individuals with MCI (Mean Age: 71.35 years, SD = 5.76). - MAP: 27 participants (20 females). - HEP: 27 participants (21 females).	Active control group	- Both MAP and HEP included weekly in-person sessions for 3 months, followed by monthly sessions for 6 months, conducted in a quiet community location. - MAP: Home practice of mindfulness techniques was encouraged. - HEP: Participants applied health education topics to daily life with provided handouts.	Community
9	Wong et al., 2017 (Australia) [55]	Longitudinal, mixed-methods observational study with a pre-/post-intervention design	- 14 older adults with MCI (8 females, Mean age: 76.5 years, SD = 6.7). - Small sample size (12 completed one-year follow-up).	No (pre-/post intervention design)	- Customized 8-week mindfulness training program for MCI patients. - Weekly 1.5-h in-person group sessions. - Participants were encouraged to practice mindfulness at home and record their practice.	Community
10	Larouche et al., 2019 (Germany) [56]	Single-blind, preliminary randomized-controlled trial (RCT)	- 44 older adults with aMCI (aged 56–87 years), randomly assigned 1:1 into two groups: - MBI group (*n* = 23) and - PBI group (*n* = 21).	Active control group	- Mindfulness-Based Intervention (MBI): Weekly 2.5-h in-person sessions for 8 weeks, with daily home meditation practice (formal and informal) and compliance tracking. Weekly phone calls were made for support. - Psychoeducation-Based Intervention (PBI): Weekly 2.5-h in-person sessions for 8 weeks, focused on psychoeducation with no home practice assigned.	Community
11	Quintana-Hernández et al., 2016 (Spain) [57]	Randomized clinical trial (RCT)	- 120 patients aged 65+ with probable AD (66 females). - MBAS: 30 participants. - CST: 30 participants. - PMR: 30 participants. - Control (Donepezil only): 30 participants. - Age distribution: 65–75 years (28 participants, 23.3%), 76–85 years (65 participants, 54.2%), ≥86 years (27 participants, 22.5%).	Pharmacological treatment-only control group	- Over a two-year period, participants in the experimental groups attended three weekly sessions of 90 min each. - The interventions included the use of donepezil and: (a) Mindfulness-based Alzheimer’s stimulation (MBAS): Inspired by mindfulness-based stress reduction (MBSR) and other mindfulness techniques. (b) Cognitive stimulation therapy (CST): A standard psychostimulation program. (c) Progressive muscle relaxation (PMR): A relaxation training program. (d) The control group received only donepezil without additional non-pharmacological treatments. - In-person	Community
12	Lim et al., 2018 (Singapore) [58]	Randomized Controlled Trial (RCT)	- Post-intervention: 36 MCI participants (17 in MAP, 19 in HEP). - Age range: 60–90 years.	Active control group	- MAP group: Taught mindfulness techniques based on McBee’s mindfulness-based elder care approach, with daily home practice and record-keeping. - HEP group: Attended educational sessions on healthy living topics, with record-keeping but no specific home exercises. - Both groups participated in weekly sessions for the first 12 weeks (40 min per session) and monthly sessions for the subsequent 6 months (45 min per session)	Community

**Table 4 brainsci-15-00580-t004:** Effectiveness of PPI interventions.

S/N	Reference	Assessment Time	Outcome Measures	Results
1	Whitfield et al., 2022 (Europe) [47]	- Baseline (week 0) - Post-intervention (week 8) - 6-month follow-up (week 24)	- Abridged Preclinical Alzheimer’s Cognitive Composite 5 (PACC5Abridged) - Trail-Making Test Part A (TMT-A) - Attention composite: Stroop Naming Condition - WAIS-IV Coding Subtest - Executive function composite: Trail-Making Test Part B (TMT-B) - Letter Fluency for “P” - Stroop Interference Score - Credibility/Expectancy Questionnaire (CEQ)	- PACC5Abridged: Significant improvements in global cognition in both groups, with no difference between groups in week 24 (not in week 8). - Attention Composite: Weak evidence of improvement, but not statistically significant. - Executive Functions: No significant improvements in either group.
2	Smart and Segalowitz, 2017 (Canada) [48]	- Pre-intervention/baseline (before 8 weeks) - Post-intervention (after 8 weeks)	- Event-Related Potentials (ERPs) - Neural Measures: EEG (ERN and Pe components) - Behavioral Measures: Eriksen Flanker Task (reaction time and accuracy) - Self-Report Measures: AMAS-E, NMR, and FFMQ (Non-Judge and Non-React subscales) - Additional Tasks: Go/No-Go Task (for other analyses)	Mindfulness Training: - Increased ERN amplitude (greater attentiveness to errors). - No reliable change in Pe amplitude. - Trend toward faster reaction times. - Psychoeducation: - Increased Pe amplitude (greater emotional reactivity to errors). - Trend toward slower reaction times. - Results are general for both MT and HC.
3	Doshi et al., 2021 (Singapore) [49]	- Pre-Intervention Assessment: Conducted within 3 months before the first intervention session. - Post-Intervention Assessment: Conducted within 3 months after the last intervention session.	- Repeatable Battery for the Assessment of Neuropsychological Status (RBANS) - Mini-Mental State Examination (MMSE) - Montreal Cognitive Assessment (MoCA) - Mindful Attention and Awareness Scale (MAAS)	- MBT showed greater improvements in delayed memory compared to CRT but no significant advantage over TAU. - Both MBT and CRT improved global cognition and visuospatial processing, but these effects were not superior to TAU. - Spontaneous cognitive improvements in the TAU group were attributed to diagnostic instability and research participation effects.
4	Ng et al., 2020 (Singapore) [50]	- Baseline: At the start of the trial, before the intervention began. - 3-Month Follow-Up: At the end of the 12-week weekly intervention phase. - 9-Month Follow-Up: At the end of the 6-month monthly booster session phase.	- Blood Samples: Used to measure hs-CRP, BDNF, and DHEA-S levels - Saliva Samples: Used to measure cortisol, IL-6, and IL-1β levels - Samples were collected at consistent times (between 9:00 and 11:00 a.m.) to minimize diurnal variations	- The MAP intervention significantly reduced hs-CRP levels at 9 months, particularly in females and participants with amnestic-MCI (aMCI), suggesting a potential anti-inflammatory effect. - Improvements in IL-6 and IL-1β were observed in males at the 3-month follow-up, but these effects were not sustained at 9 months. - No significant changes were observed in BDNF, cortisol, or DHEA-S levels.
5	Ng et al., 2022 (Singapore) [51]	- Baseline (prior to the intervention) - 3 months (immediately after the weekly intervention sessions) - 9 months (after the monthly booster sessions) - 5 years (long-term follow-up)	- Mini-Mental State Examination (MMSE)	- MAP did not significantly delay cognitive decline compared to HEP.
6	Fam et al., 2020 (Singapore) [52]	- Baseline: Before the 3-month intervention began. - Post-intervention: After the 3-month intervention was completed	- Modified Mini-Mental State Examination (MMSE) - Rey Auditory Verbal Learning Test (RAVLT) - Block Design Test (from the Wechsler Adult Intelligence Scale) - functional Magnetic Resonance Imaging (fMRI)	- Cognitive Function: Significant improvement in recognition memory in the mindfulness group compared to controls. - Brain Connectivity: - Global Efficiency: Preserved in the mindfulness group, while it declined in the control group. - Local Efficiency: Improved over time in both groups. - Nodal Efficiency: Improved in specific brain regions in the mindfulness group, including the right insula, right cingulate gyrus, and left superior temporal gyrus.
7	Klainin-Yobas et al., 2019 (Singapore) [53]	- Baseline (Time 1): Within 1–2 weeks before the start of the intervention. - Three months (Time 2): After the completion of weekly sessions. - Nine months (Time 3): After the completion of monthly sessions	- Modified Mini-Mental State Examination (MMSE) - Clinical Dementia Rating (CDR)	- Cognitive Function (MMSE and CDR): No significant improvement was observed in either group over the 9-month period.
8	Yu et al., 2021 (Singapore) [54]	- Baseline (prior to the intervention). - 3 months (after the weekly sessions phase). - 9 months (after the monthly sessions phase).	- Semantic fluency test - Color trails test - Digit span subtests (WAIS-III) - Rey Auditory Verbal Learning Test - Block design test (WAIS-III) - MRI - CAT12 toolbox (version r1450)	Mindfulness Awareness Program (MAP) led to: - Cognitive Improvements: Significant gains in working memory span and divided attention at 9 months. Cortical Thickness Changes: - Increased cortical thickness in the right frontal pole at 9 months. - Decreased cortical thickness in the left anterior cingulate cortex at 9 months. - Temporary increase in the left inferior temporal gyrus at 3 months, not sustained at 9 months.
9	Wong et al., 2017 (Australia) [55]	- Pre-intervention (T1): Two weeks before the intervention. - Post-intervention (T2): Two weeks after the eight-week mindfulness program. - One-year follow-up (T3): 59 weeks after the pre-intervention assessment.	- Montreal Cognitive Assessment (MoCA) - 14-item Freiburg Mindfulness Inventory (FMI) - 12-item Mindfulness Adherence Questionnaire (MAQ) - 25-item Bayer Activities of Daily Living Scale (B-ADL) - Demographic, Health, and Lifestyle (DHL) questionnaire	- Significant improvement in cognitive function even at T3. - Trait mindfulness significantly improved. - Improvements in ADL functioning.
10	Larouche et al., 2019 (Germany) [56]	- Baseline (T0): Pre-intervention assessment conducted before the start of the intervention. - Post-intervention (T1): Conducted one week after the completion of the 8-week intervention. - Follow-up (T2): Conducted three months after the post-intervention assessment.	- World Health Organization Quality of Life Brief Scale (WHOQOL-BREF) - World Health Organization Quality of Life Old Scale (WHOQOL-OLD) - Free recall verbal episodic memory task - Five-Facet Mindfulness Questionnaire (FFMQ) - Ruminative Response Scale (RRS)	- Both interventions (MBI and PBI) were effective in improving aging-related quality of life. - Neither intervention improved general quality of life or episodic memory. - The MBI showed unique mechanisms of change, with non-judgment and rumination reduction playing a role in its psychological benefits.
11	Quintana-Hernández et al., 2016 (Spain) [57]	- Baseline (Month 0) - 6 months - 12 months - 18 months - 24 months (end of the study)	- CAMDEX-R: The Cambridge Examination for Mental Disorders of the Elderly—Revised version, which includes: - Mini-Mental State Examination (MMSE) - Cambridge Cognitive Examination (CAMCOG)	- MBAS: Most effective in maintaining cognitive capacities across all domains (memory, attention, language, praxis) over 2 years. Equivalent to CST but superior to PMR and control. - CST: Effective but less stable after 12 months. Superior to PMR and control. - PMR: Moderate initial effect, declined over time. It is not significantly different from control in most domains. - Control: Most rapid cognitive decline.
12	Lim et al., 2018 (Singapore) [58]	- Pre-Intervention: Blood samples were collected from participants before the start of the intervention (baseline assessment). - Post-Intervention: Blood samples were collected after nine months of intervention (final assessment)	- Affymetrix Human Genome U133 Plus 2.0 arrays - DAVID bioinformatics tool - Genetic Association Database (GAD) and Online Mendelian Inheritance in Man (OMIM)	- MAP (Mindfulness Awareness Practice): Changes in gene expression, with enrichment in pathways related to neuroactive ligand-receptor interactions and stress-related pathways. - HEP (Health Education Program): Changes in gene expression, with enrichment in pathways associated with metabolism and cellular signaling.

**Table 5 brainsci-15-00580-t005:** Cognitive and brain functioning outcomes of mindfulness-based interventions.

Intervention	Cognitive Functions	Brain Functioning
MAP (Mindful Awareness Practice)	- Short-term improvement in recognition memory, working memory span, and divided attention. - Long-term effects are limited and inconsistent.	- Enrichment of biological pathways (immune response, inflammation, cellular stress). - Reduction of inflammatory biomarkers - Temporary changes in cortical thickness in the left inferior temporal gyrus at 3 months.
MBT (Mindfulness-Based Training)	- Improvement in global cognition and delayed memory. - Not superior to control groups.	Not reported.
MT (Mindfulness Training)	- No significant improvement in memory. - Improvement in error monitoring (attentiveness to errors).	- Increased ERN (Error-Related Negativity) amplitude, indicating better error monitoring.
MTP (Mindfulness Training Program)	- Significant and lasting improvement in cognitive functions and daily living functioning, effects maintained up to one-year post-intervention.	Not reported.
MBI (Mindfulness-Based Intervention)	- No improvement in episodic memory. - Improved aging-related quality of life	Not reported.
MBAS (Mindfulness-Based Alzheimer’s Stimulation)	- Maintenance of cognitive abilities (memory, attention, language, praxis) over two years.	Not reported.
CMBAS (Caring Mindfulness-Based Approach for Seniors)	- Improvement in global cognition but not superior to the Health Self-Management Program (HSMP) group. - Weak evidence for improvement in attention. - No improvement in executive function.	Not reported.

**Table 6 brainsci-15-00580-t006:** Quality appraisal.

A. Quantitative Randomized Controlled Trials (*n* = 11)											
Study		2.1. Is Randomization Appropriately Performed?		2.2. Are the Groups Comparable at the Baseline?		2.3. Are There Complete Outcome Data?		2.4. Are Outcome Assessors Blinded to the Intervention Provided?		2.5 Did the Participants Adhere to the Assigned Intervention?	
Yu et al. (2021) [54]		Yes		Yes		No		No		Yes	
Klainin-Yobas et al. (2019) [53]		Yes		Yes		No		No		Can’t tell	
Larouche et al. (2019) [56]		Yes		Yes		No		Yes		Yes	
Doshi et al. (2021) [49]		Yes		Yes		No		Yes		Yes	
Ng et al. (2022) [51]		Yes		Yes		No		Yes		Yes	
Fam et al. (2020) [52]		Yes		Yes		No		Can’t tell		Yes	
Quintana-Hernández et al. (2015) [57]		Yes		Yes		No		No		Yes	
Whitfield et al. (2022) [47]		Yes		Yes		Yes		Yes		Yes	
Smart and Segalowitz (2017) [48]		Yes		Yes		No		Yes		Yes	
Ng et al. (2020) [50]		Yes		Yes		No		Yes		Can’t tell	
Lim et al. (2018) [58]		Yes		Can’t tell		No		Can’t tell		Can’t tell	
**B. Quantitative Descriptive and Mixed Methods Study (*n* = 1)**											
**Study**	**4.1**	**4.2**	**4.3**	**4.4**	**4.5**	**5.1**	**5.2**	**5.3**	**5.4**	**5.5**	
Wong et al. (2017) [55]	Yes	No	Yes	No	Yes	Yes	Can’t tell	Can’t tell	Can’t tell	Yes	
Note. The criteria used are listed below due to space limitations in the table.											
4.1. Is the sampling strategy relevant to address the research question? 4.2. Is the sample representative of the target population? 4.3. Are the measurements appropriate? 4.4. Is the risk of nonresponse bias low? 4.5. Is the statistical analysis appropriate to answer the research question? 5.1. Is there an adequate rationale for using a mixed methods design to address the research question? 5.2. Are the different components of the study effectively integrated to answer the research question? 5.3. Are the outputs of the integration of qualitative and quantitative components adequately interpreted? 5.4. Are divergences and inconsistencies between quantitative and qualitative results adequately addressed? 5.5. Do the different components of the study adhere to the quality criteria of each tradition of the methods involved?											

## Data Availability

No new data were created or analyzed in this study.

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
