# Peer review of "Positive Psychology Interventions in Early-Stage Cognitive Decline Related to Dementia: A Systematic Review of Cognitive and Brain Functioning Outcomes of Mindfulness Interventions"

_brainsci, 2025, doi:10.3390/brainsci15060580_

Round 1
Reviewer 1 Report
Comments and Suggestions for Authors
This systematic review addresses a highly relevant topic; however, it presents methodological weaknesses that compromise the robustness and reproducibility of its findings. The following are the main aspects that require revision and improvement:
Introduction
The introduction lacks a clearer theoretical framework regarding existing evidence in the literature and the specific gap this systematic review aims to address. Differences between the included studies in terms of interventions and outcomes are not clearly discussed. Additionally, the objective of the review is not scientifically well-justified, especially considering that no statistical analysis or meta-analysis was performed, which limits the ability to assess the effectiveness of the reviewed interventions. It is therefore recommended to reformulate the review’s objective to better reflect its actual scope and methodological approach.
Methods
It is recommended that the review be registered on the PROSPERO platform to enhance transparency and credibility.
The inclusion and exclusion criteria should be described in greater detail and presented in a narrative format rather than bullet points, improving scientific rigor and coherence.
The search strategy is not sufficiently described to ensure replicability. As each database has specificities (e.g., searching by title, keywords, or abstract), it is essential to include the Boolean operators used for each database, ensuring clarity on how descriptors were applied.
The PRISMA flowchart contains inconsistencies: the identification of records by each database is not specified, and reasons for article exclusion are not listed in the diagram, with only around 64 articles addressed in Table 1.
Additionally, key methodological details are missing, such as the subitems:
- The procedure for data extraction from the included studies;
- The tool used to assess the risk of bias;
- The method applied to analyze the extracted data.
Results
Figure 1 must be self-explanatory and include a clear legend with all symbols used, including an explanation of the asterisks, which currently lack clarification.
It would be valuable to label the 311 excluded articles with their reasons for exclusion and indicate which databases contributed most to the exclusions. This would strengthen the transparency of the selection process and guide future reviews in the field.
Although the results are described adequately, the tables are overly condensed, making it difficult to interpret the data. It is recommended to divide the tables by context or category, improving clarity and aligning better with the text narrative.
Discussion and Conclusion
The discussion and conclusion are consistent with the results presented. However, the methodological adjustments suggested above will enhance the scientific support for the discussion points and reinforce the conclusions.
Author Response
Introduction
Comment 1: The introduction lacks a clearer theoretical framework regarding existing evidence in the literature and the specific gap this systematic review aims to address. Differences between the included studies in terms of interventions and outcomes are not clearly discussed. Additionally, the objective of the review is not scientifically well-justified, especially considering that no statistical analysis or meta-analysis was performed, which limits the ability to assess the effectiveness of the reviewed interventions. It is therefore recommended to reformulate the review’s objective to better reflect its actual scope and methodological approach.
Response 1: We have added a paragraph at the beginning of the article that elaborates on the objective of our research in greater detail. Literature suggests correlations between various constructs of positive psychology and the enhancement of cognitive and brain functioning; therefore, they could potentially serve as non-pharmacological interventions for dementia. The aim of our systematic review was to investigate whether this idea has already been implemented in practice and to examine the results. However, the findings are scarce and limited to the application of mindfulness only, rather than other elements of positive psychology. The systematic review identified only 12 studies, which we consider insufficient for conducting a meta-analysis. Nevertheless, we have added a 'Quality Appraisal' (3.7) section at the end of the 'Results' section. Finally, the differentiation in interventions is explained in more detail in section 3.6.
Methods
Comment 2: It is recommended that the review be registered on the PROSPERO platform to enhance transparency and credibility
Response 2: Yes for submission, but the time is limited.
Comment 3: The inclusion and exclusion criteria should be described in greater detail and presented in a narrative format rather than bullet points, improving scientific rigor and coherence.
Response 3: Regarding the inclusion and exclusion criteria, we have changed their presentation to a narrative format. As for the content, we believe that they are complete and clear.
Comment 4: The search strategy is not sufficiently described to ensure replicability. As each database has specificities (e.g., searching by title, keywords, or abstract), it is essential to include the Boolean operators used for each database, ensuring clarity on how descriptors were applied.
Response 4: Regarding the search strategy for each database, we have added it to the Appendix F for each of the four databases.
Comment 5: The PRISMA flowchart contains inconsistencies: the identification of records by each database is not specified, and reasons for article exclusion are not listed in the diagram, with only around 64 articles addressed in Table 1.
Response 5: We believe that the PRISMA diagram is sufficiently clear based on the original template. Table 1 lists the articles that we read in full after the initial screening, during which exclusions were made. We found 887 articles, and after removing duplicates (500), 387 remained, of which we excluded 311. These 311 articles were excluded during the screening phase because, based on their titles and abstracts, the reviewers determined that they did not meet the inclusion criteria. This screening process was conducted manually by the reviewers, without the use of any automated tool. Subsequently, we read the remaining 76 articles in detail, and in Table 1, we specify the reasons for excluding 64 and ultimately including 12.
Comment 6: The procedure for data extraction from the included studies;
Response 6: We have addressed your comment regarding the data extraction procedure by adding the following information to section 2.5: “Data extraction was conducted using a pre-defined data extraction form developed by the authors. The following data elements were extracted from each included study: reference (author, year), design of the study (e.g., RCT, pre-post intervention), population (characteristics of participants, including cognitive status), control group (presence or absence of a control group and type of control group), PPI (type of intervention, duration, frequency), assessment time (time points of assessment), outcome measures (cognitive and brain functioning measures), results (main findings related to cognitive and brain functioning), and the setting where participants were recruited from (community or hospital). Two independent reviewers extracted data, and discrepancies were resolved by consensus.” We believe this addition clarifies our data extraction process.
Comment 7: The tool used to assess the risk of bias;
Response 7: Regarding the need for information on the risk of bias assessment, we have addressed this comment by adding a new section, 3.7, titled 'Quality Appraisal'. This section describes the tool used and summarizes the results of the appraisal. Also the procedure is in Table 5.
Comment 8: The method applied to analyze the extracted data.
Response 8: We have addressed your comment regarding the method applied to analyze the extracted data by adding the following information to section 2.6, 'Synthesis methods': “ Due to the limited number of studies a quantitative meta-analysis was not possible. Therefore, we conducted a qualitative synthesis of the data, summarizing the key findings and identifying common themes across studies.
Results
Comment 9: Figure 1 must be self-explanatory and include a clear legend with all symbols used, including an explanation of the asterisks, which currently lack clarification.
Response 9: About your feedback on Figure 1. We have revised the figure to include a clear legend explaining all symbols used, including the asterisks, to ensure it is self-explanatory. We believe this addition improves the clarity and understandability of the figure.
Comment 10: It would be valuable to label the 311 excluded articles with their reasons for exclusion and indicate which databases contributed most to the exclusions. This would strengthen the transparency of the selection process and guide future reviews in the field.
Response 10: Regarding your suggestion to provide more detail on the excluded articles, we describe the exclusion process in section 3.1. As stated there, 311 articles were excluded during the title and abstract screening phase because they did not meet the inclusion criteria. The specific reasons for excluding the 64 full-text articles are detailed in Table 1. Due to the large number of articles excluded during the initial screening, providing a detailed breakdown of exclusion reasons and contributing databases for those 311 articles would be impractical. However, we believe the information provided in section 3.1 and Table 1 adequately clarifies the selection process.
Comment 11: Although the results are described adequately, the tables are overly condensed, making it difficult to interpret the data. It is recommended to divide the tables by context or category, improving clarity and aligning better with the text narrative.
Response 11: We have addressed your concern about the tables being overly condensed by dividing Table 2 into two separate tables: Table 2 and Table 3. Table 2 now presents the characteristics of the interventions, while Table 3 presents their effectiveness. We believe this separation improves the clarity and readability of the data, and better aligns with the narrative in the text.
Reviewer 2 Report
Comments and Suggestions for Authors
The authors presented a systematic review of the effectiveness of positive psychology intervention for patients with dementia. The authors use the generally accepted PRISMA method as the basis of their review, in addition, they strictly formulate the criteria for including and excluding articles in the review, and additional materials include descriptions of each article with the reasons for inclusion/exclusion. As a result, 12 publications are analyzed, in which almost seven hundred participants were considered. Mindfulness-based interventions were used as the main types of impacts. The approaches considered in this direction have shown a positive impact, but the review has not revealed long-term effectiveness (the authors present the limitations of the articles reviewed). The article includes a fairly large discussion section with a comprehensive analysis of the reviewed papers, the prospects for various intervention methods and their limitations. Thus, the presented systematic review meets all the requirements for this class of articles and can be considered for publication.
After studying, I still have the following questions and suggestions:
1. Perhaps the authors should slightly adjust the title, since of all the areas of positive psychology they consider only interventions aimed at mindfulness, therefore, "A systematic review of cognitive and brain functioning outcomes of Mindfulness Interventions in early-stage cognitive decline related to dementia" would be more logical. Or something else like that.
2. Given the relatively close topics of the articles under consideration, if possible, it would be worthwhile to add a comparison of quantitative metrics and the quantitative effect of the results obtained in the articles (meta-analysis). I do not limit the authors in the choice of methods/ ways of demonstrating it, and I will also accept if this aspect is difficult due to the high complexity.
3. All the included studies were conducted in developed countries, and I did not see any factor limiting articles from other countries in the exclusion criteria (with the exception of the language of writing). Because I don't really understand the phrase in table 1 "Other population". Please comment on whether there were suitable studies in developing countries, and if not, by what criteria they did not fit into this review.
4. Perhaps the Limitations section (or a separate section) should also include criticism and limitations of Mindfulness Interventions, an overview of the requirements for their application, for example, restrictions on the resources of patients or hospitals.
Author Response
Comment 1: Perhaps the authors should slightly adjust the title, since of all the areas of positive psychology they consider only interventions aimed at mindfulness, therefore, "A systematic review of cognitive and brain functioning outcomes of Mindfulness Interventions in early-stage cognitive decline related to dementia" would be more logical. Or something else like that.
Response 1: We decided to use this title because our aim was to investigate every positive psychology intervention that has been administered for this purpose (that is, to enhance cognitive and brain functioning). However, through our study, we found that only mindfulness-based interventions have been administered so far. This highlights a significant gap in the literature. We have decided to keep the title as it is because we want to emphasize exactly this point: that the purpose of the systematic review was to identify all such interventions, not just those focused on mindfulness. The fact that we found only mindfulness-based interventions is indicative of a lack of research and an emerging, innovative scientific field.
Comment 2: Given the relatively close topics of the articles under consideration, if possible, it would be worthwhile to add a comparison of quantitative metrics and the quantitative effect of the results obtained in the articles (meta-analysis). I do not limit the authors in the choice of methods/ ways of demonstrating it, and I will also accept if this aspect is difficult due to the high complexity.
Response 2: Certainly, a meta-analysis would be helpful, but the systematic review identified only 12 studies, which we consider insufficient for conducting a meta-analysis. Nevertheless, we have added a 'Quality Appraisal' (3.7) section at the end of the 'Results' section
Comment 3: All the included studies were conducted in developed countries, and I did not see any factor limiting articles from other countries in the exclusion criteria (with the exception of the language of writing). Because I don't really understand the phrase in table 1 "Other population". Please comment on whether there were suitable studies in developing countries, and if not, by what criteria they did not fit into this review.
Response 3: There was no restriction regarding the country, only the filter that the articles be written in English. This fact is obviously due to the very limited data available so far. Regarding "other population," we mean studies that do not include people with SCD (Subjective Cognitive Decline), MCI (Mild Cognitive Impairment), and mild AD (Alzheimer's Disease) dementia. We have added this as an explanation to Table 1 and Figure 1 where they are mentioned
Comment 4: Perhaps the Limitations section (or a separate section) should also include criticism and limitations of Mindfulness Interventions, an overview of the requirements for their application, for example, restrictions on the resources of patients or hospitals
Response 4: We added an additional paragraph in the limitations section to highlight the challenges of implementing mindfulness interventions. We also included a new category in Table 2 regarding whether the participants in the studies were recruited from the community or from hospitals/institutions. As you can see, participants came from both settings, which indicates that these interventions can be delivered in both cases. However, in the suggestions for future research, we added: "Finally, these interventions could also be delivered online to be more accessible to everyone, regardless of their location.
Reviewer 3 Report
Comments and Suggestions for Authors
The topic addressed in this systematic review is highly relevant, especially considering the growing interest in non-pharmacological interventions for cognitive decline. The authors provide a comprehensive summary of mindfulness-based interventions and their impact on cognitive and brain functioning in SCD, MCI, and early AD. However, some improvements are necessary:
-
Clarity of Results Presentation: While the synthesis is detailed, the organization of the results (especially in section 3.6) is dense and could benefit from summarizing key findings in clearer comparative tables or bullet points to enhance readability.
-
Language and Style: Several sections contain awkward phrasing or overly long sentences, which affect the clarity of the manuscript. A thorough proofreading by a native English speaker is recommended to improve fluency and readability.
-
Figures and Visual Summaries: A summary table or figure comparing cognitive domains improved (or not) across the different interventions would greatly enhance the reader's understanding.
-
Discussion Depth: While the discussion is well-informed, it would benefit from a clearer distinction between confirmed effects and preliminary findings, with more emphasis on methodological heterogeneity and limitations in study design (e.g., sample sizes, different biomarkers measured, lack of blinding).
-
Character Strengths: Although the introduction sets up a broader framework that includes other positive psychology constructs (e.g., character strengths), the review ends up focusing almost exclusively on mindfulness. The authors might want to either adjust the framing or more clearly state that this reflects a gap in the literature.
The manuscript requires careful revision for English language clarity. While the scientific content is understandable, the text contains several grammatical errors, awkward constructions, and overly long or complex sentences that reduce readability. A thorough review by a native English speaker or professional language editing service is strongly recommended to improve fluency and ensure clearer communication of the research findings.
Author Response
Comment 1: Clarity of Results Presentation: While the synthesis is detailed, the organization of the results (especially in section 3.6) is dense and could benefit from summarizing key findings in clearer comparative tables or bullet points to enhance readability.
Response 1: We have split Table 2 into Table 2 and Table 3 to improve readability and clarity. Table 3 now presents the results of each study. We also added Table 4. Cognitive and brain functioning outcomes of mindfulness-based interventions.
Comment 2: Language and Style: Several sections contain awkward phrasing or overly long sentences, which affect the clarity of the manuscript. A thorough proofreading by a native English speaker is recommended to improve fluency and readability.
Response 2: The manuscript has been reviewed and edited by a professor of English literature to improve fluency and readability. We believe the current version reflects those improvements.
Comment 3: Figures and Visual Summaries: A summary table or figure comparing cognitive domains improved (or not) across the different interventions would greatly enhance the reader's understanding.
Response 3: We added Table 4. Cognitive and brain functioning outcomes of mindfulness-based interventions.
Comment 4: Discussion Depth: While the discussion is well-informed, it would benefit from a clearer distinction between confirmed effects and preliminary findings, with more emphasis on methodological heterogeneity and limitations in study design (e.g., sample sizes, different biomarkers measured, lack of blinding).
Response 4: We added a paragraph to highlight these issues in the discussion and one in the limitations section.
Comment 5: Character Strengths: Although the introduction sets up a broader framework that includes other positive psychology constructs (e.g., character strengths), the review ends up focusing almost exclusively on mindfulness. The authors might want to either adjust the framing or more clearly state that this reflects a gap in the literature.
Response 5: We added the following sentence to the end of the discussion's first paragraph: “ Our aim was to investigate every study that uses different constructs of positive psychology to enhance cognitive functions and brain functioning in individuals with cognitive decline related to dementia. However, our systematic review focuses on mindfulness, as it appears that, so far, only such studies have been conducted. This highlights a significant gap in the literature.”
Round 2
Reviewer 3 Report
Comments and Suggestions for Authors
The revisions made to the manuscript have significantly improved the methodological clarity, the argumentative structure, and the alignment with the aims of the review.
In particular, I appreciate:
- the clearer articulation of the inclusion criteria and the study selection process;
- the reformulation of certain statements concerning the still preliminary nature of the effectiveness of mindfulness-based interventions;
- the expanded critical discussion in the “Discussion” section, especially regarding methodological limitations (e.g., heterogeneity of protocols, small sample sizes, limited follow-up periods). In light of these revisions, I consider the manuscript ready for publication. Although preliminary, the evidence gathered offers a valuable and up-to-date contribution to the literature on the role of Positive Psychology Interventions in early-stage cognitive decline. The paper strikes a good balance between scientific rigor and clinical relevance, and meets the standards expected by the Neuropsychology section.